# The Skin Microbiome in Cutaneous T-Cell Lymphomas (CTCL)—A Narrative Review

**DOI:** 10.3390/pathogens11080935

**Published:** 2022-08-18

**Authors:** Magdalena Łyko, Alina Jankowska-Konsur

**Affiliations:** Department of Dermatology, Venereology and Allergology, Wrocław Medical University, 50-368 Wrocław, Poland

**Keywords:** cutaneous T-cell lymphoma, mycosis fungoides, Sézary syndrome, skin microbiome, microbiota, *Staphylococcus aureus*, treatment

## Abstract

In recent years, numerous studies have shown a significant role of the skin microbiome in the development and exacerbation of skin diseases. Cutaneous T-cell lymphomas (CTCL) are a group of malignancies primary involving skin, with unclear pathogenesis and etiology. As external triggers appear to contribute to chronic skin inflammation and the malignant transformation of T-cells, some microorganisms or dysbiosis may be involved in these processes. Recently, studies analyzing the skin microbiome composition and diversity have been willingly conducted in CTCL patients. In this review, we summarize currently available data on the skin microbiome in CTLC. We refer to a healthy skin microbiome and the contribution of microorganisms in the pathogenesis and progression of other skin diseases, focusing on atopic dermatitis and its similarities to CTCL. Moreover, we present information about the possible role of identified microorganisms in CTCL development and progression. Additionally, we summarize information about the involvement of *Staphylococcus aureus* in CTCL pathogenesis. This article also presents therapeutic options used in CTCL and discusses how they may influence the microbiome.

## 1. Introduction

Cutaneous T-cell lymphomas (CTCL) are a heterogeneous group of neoplasms with the primary presentation in the skin. Mycosis fungoides (MF) and its leukemic counterpart, Sézary syndrome (SS), are the most common subtypes. MF typically presents with erythematous patches, plaques, tumors, or generalized erythroderma, accompanied by scaling and pruritus. The course is indolent, and the prognosis in the early stages is excellent [1,2,3]. In advanced stages, lymph nodes are most likely affected. However, extracutaneous disease may involve any organ of the body [4,5]. SS more often presents as erythroderma and generalized lymphadenopathy that develops over weeks to months. Compared to MF, the disease has an aggressive course and poor prognosis [1,2,3]. In Europe and the USA, the incidence of CTCL is estimated from 0.55 to 1.06 per 100,000 persons [6,7,8,9]. MF constitutes over half of these cases, reaching an incidence of up to 0.56 per 100,000 persons [10]. SS incidence is estimated at 0.075 per 100,000 persons [11].

## 2. Pathogenesis and Link with Microorganisms

The pathogenesis of CTCL is not fully understood, but many mechanisms seem to play a part. Both MF and SS develop from skin-homing CD4+ T-cells. However, different subsets of memory T-cells are observed in MF and SS. MF originates from the skin resident memory T-cell (TRM), whereas SS comes from the skin-tropic central memory T-cell (TCM) [12]. These malignant T-cells most often express cutaneous lymphocyte antigen (CLA), CC chemokine receptor 4 (CCR4), and CC chemokine receptor (CCR10) [13,14,15,16,17]. In these lymphoproliferative disorders, an impaired immune response is associated with more frequent infections and suppression of the anti-tumor reactions [18]. Among others, external factors have been proposed as the ones related to the development and/or exacerbation of the disease [19]. A recent hypothesis suggests the role of external antigens that contribute to chronic inflammation, resulting in malignant transformation [18,19,20]. An imbalance of microorganisms may trigger the innate immune system, causing the mentioned relationship [21]. This has been described in atopic dermatitis, acne vulgaris, psoriasis, and others [22,23,24,25]. Considering the above, dysbiosis may impact the course or promote the initiation of CTCL. With the development of modern technologies, especially genetic methods such as DNA sequencing, researchers more willingly study the topic of the human skin microbiome in CTCL. This review summarizes the knowledge about the skin microbiome in health and disease. In particular, we will discuss recent insights into the skin microbiome in CTCL, focusing on research using sequencing technology and molecular diagnostic methods.

## 3. Healthy Skin Microbiome

The microbiome is a term describing diverse microorganisms, such as bacteria, viruses, fungi, and arthropods, that inhabit different parts of the human body, including the skin, gut, oral cavity, and nostrils [26,27,28]. Focusing on the skin microbiome, in most skin areas, bacteria constitute over 70% of the microbiota, whereas fungi are the least abundant of all microbes [28,29]. Four bacteria phyla, *Actinobacteria*, *Firmicutes*, *Bacteroidetes*, and *Proteobacteria*, are the most prevalent on the human skin. Moreover, fungi and mites from the genera, *Malassezia* spp. and *Demodex* spp., are skin residents. Viruses constitute an essential part of the microbiome. However, they are an unstable component of the skin habitat. In skin microbiome research, double-stranded DNA (ds. DNA) viruses, particularly *Polyomaviridae* and *Papillomaviridae*, are frequently reported [30,31]. Beneficial interactions are observed between the host and the microbiota. The commensal microorganisms on the skin participate in the process of maturation of the host immune system, and provide homeostasis of cutaneous immunity [32,33,34,35]. Moreover, certain bacterial species metabolize lipids on the skin and stimulate or produce antimicrobial peptides (AMPs), which protect against invasion by more pathogenic microbes [36,37,38,39]. In exchange, the skin distinguishes antigens of residual microorganisms, and no inflammatory processes take place in response to them [28,32,40]. However, in specific settings, microbes can exhibit pathogenic potential, exacerbating skin lesions, promoting disease development, and delaying wound-healing [41].

Chemical and physical characteristics of the skin vary in different regions of the human body. The density and variety of sweat and sebaceous glands, and hair follicles are distinct in different body areas. What is more, the topography has an impact on temperature and moisture, which affect microorganism colonization [42]. Accordingly, human skin can be divided into particular niches, such as moist (the axilla, inner elbow, or inguinal fold), dry (the volar forearm, the abdomen, the upper buttock area), sebaceous (the forehead, the alar crease, the retro auricular crease, and the back), and others [30,42,43]. Among others, the frequently mentioned foot microbiome is the most unique and heterogeneous niche. It is relatively unstable and is characterized by a more diverse fungi composition not shown in any other body site [28]. Moreover, factors such as genotype, age, and sex, as well as geographical location, occupation, lifestyle, and the use of antibiotics or cosmetics, may affect pH, moisture, salinity, and sebum content. All of the above factors explain why the diversity of the skin microbiota differs in different regions of the body and between individuals [30,32,44,45,46,47]. Moist skin is characterized by a higher abundance of *Firmicutes* (*Staphylococcus* spp.) and *Actinobacteria* (*Micrococcus* spp., *Corynebacterium* spp.), and sebaceous regions are characterized by *Cutibacterium* spp. presence. Dry skin is dominated by *Actinobacteria*, *Firmicutes*, *Bacteroidetes*, and *Proteobacteria*. Fungal composition, contrary to bacterial communities, is predominated by *Malassezia* spp., regardless of the body site. Only foot areas show varying diversity with the presence of *Malassezia* spp., *Cryptococcus* spp., *Aspergillus* spp., and others [43,48]. This body site dependence is not observed in eukaryotic DNA viruses, and they are more specific for each individual rather than for a body site [49].

## 4. The Role of the Microbes in Skin Diseases Development and Progression

The imbalance of the skin microbiome is observed in several skin diseases. However, the discussion is still going on whether there is a causal or an effect relationship between dysbiosis and the development of inflammatory skin disorders [25,50,51,52]. Two models presenting mechanisms driven by host pathologies and microbial communities were introduced [53]. The role of dysbiosis was described in several skin diseases, such as atopic dermatitis, psoriasis, acne vulgaris, hidradenitis suppurativa, and others [25,51,53,54,55,56]. Shifts in the composition of the skin microbiome are observed in such situations as the process of inflammation, tissue repair, skin barrier dysfunction, and treatment [32,57].

### The Impact of Microbes on the Course of Atopic Dermatitis

Atopic dermatitis (AD) is a common, chronic, relapsing inflammatory skin disorder associated with atopic manifestations, such as allergic rhinitis, asthma, and food allergies [58]. AD, especially with the onset of advanced age, may mimic CTCL and cause diagnostic difficulties. There are several similarities between CTCL and AD. In patients with CTCL, skin and nasal colonization of *Staphylococcus aureus* is similar to that in AD, and significantly higher than in the general population [59]. Again, similarly to AD, clinical improvement in CTCL is associated with *S. aureus* eradication [59,60]. In AD, numerous studies documented the colonization of *S. aureus* and its relation to skin barrier dysfunction. This creates the vicious circle where staphylococcal extracellular proteases break down the epidermal permeability barrier, which causes disease progression [61]. The compromised skin barrier in AD promotes the colonization of bacteria, generating higher transepidermal water loss (TEWL), resulting in increased bacterial colonization. In addition, in the lesional skin of CTCL, increased TEWL was confirmed, which is in line with the mentioned observations [62,63]. Another factor causing barrier dysfunction and, hence, higher TEWL, is scratching due to pruritus in AD and CTCL [64].

Many studies have focused on *S. aureus*; however, the possible role of another dominant skin resident has been reported. *Corynebacterium* spp. overexpression was also observed in AD patients. A higher abundance of *Corynebacterium spp.*, *S. aureus*, and *Clostridiales* correlated with AD severity [22]. In addition, skin inflammation can also be driven by *Corynebacterium accolens*. A recent study focused on the role of the mentioned bacteria in inflammatory processes in the skin. *C. accolens* promoted IL-23 signaling and activated γδ T-cells that favor skin inflammation [65].

Moreover, interactions between microbes influence human health. Skin-resident microbiomes compete for the area with each other and with potential pathogens [32,39]. Microbes can act as pathobionts, pathogens, or mutualists in certain cases. *Staphylococcus* spp.—to be more precise, *S. epidermidis* and *S. hominis*—secrete peptides that suppress the growth and even kill *S. aureus* [38,66,67]. Moreover, lantibiotics, a group of antibiotic-like peptides mainly produced by the *Staphylococcus* genus, may inhibit the growth of other bacteria [68]. For example, cytoplasmic bacteriocins produced by *S. epidermidis* showed antimicrobial activity against *S. aureus* in vitro [69]. In AD, *S. aureus* has an adverse impact on the course of the disease. The antimicrobial peptides secreted by *S. epidermidis* benefit the host, resulting in *S. aureus* eradication [32,39]. However, in AD patients, the overabundance of *S. epidermidis* may cause the expression of a cysteine protease, EcpA. The cysteine protease, EcpA, interacts with the skin immune system and exacerbates the disease course similarly to *S. aureus* [70]. This shows that in the case of microbiome disbalance, not only *S. aureus*, but also *S. epidermidis*, may be responsible for AD flares. A proper immunological response is compromised due to bacterial dysbiosis and reduction in the commensal population [33,35,71]. Interactions between *S. aureus* and *Corynebacterium* spp. have also been reported. There are many similarities between the microbiota of epithelial surfaces of the nasal passages and skin. The inhibitory activity of *C. accolens* isolated from the nasal cavity was noted in relation to *S. aureus* [72]. Moreover, the influence of *C. striatum* on *S. aureus* has been shown to result in the loss of expression of *S. aureus* virulence factors [73].

## 5. The Skin Microbiome and Role of Microorganisms in CTCL

In CTCL development, T-cell activation and the transformation to malignant forms play a crucial role. Among many factors in this pathway, antigen triggers were alleged to be involved in pathogenesis [74]. Hence, the role of microorganisms was considered one of the causes of CTCL. To determine the role of microbes, a STAT3-driven mouse model of CTCL was used. Mice were split into two groups and placed in standard and germ-free conditions after birth. Both groups developed CTCL, but the disease severity in the population set in a germ-free environment was less pronounced [75]. This may indicate that some microorganisms or dysbiosis are involved in CTCL progression.

### 5.1. The Skin Microbiome and CTCL

Recently, authors very willingly investigated microbiome diversity in CTCL. The first research published by Salava et al. [76] studied the skin microbiome in 20 patients diagnosed with MF, stage IA-IIB. The control samples were collected from contralateral healthy-looking skin of the same individual. However, the authors did not show significant differences in microbial diversity or at the genus level. In WGS data analysis, they observed a higher abundance of *Staphylococcus argenteus* in lesional skin, but after additional investigation, this observation was not noticed (Table 1.). However, the authors detected ten bacterial species (*Streptomyces* sp. *SM17*, *Bordetella pertussis*, *Streptomyces* sp. *PVA 94-07*, *Methylobacterium oryzae*, *Serratia* sp. *LS-1*, *Burkholderia mallei*, *Enterobacteriaceae bacterium*, *Achromobacter ruhlandii*, *Pseudomonas* sp. *A214*, *Pseudomonas* sp. *st29*) that were more abundant in non-lesional skin (Figure 1.). There is a possibility that the reduction of these bacteria plays a role in CTCL development. Still, as most subsequent studies did not focus on or collect samples from healthy-appearing skin, there are not enough data to make a conclusion. Another study led by Salava et al. [77], considering the skin microbiome in plaque parapsoriasis, is worth mentioning. Parapsoriasis, especially a large plaque entity, may be clinically indistinguishable from MF. Moreover, in some cases, progression to MF is observed [78,79]. In the mentioned microbiome study, 13 parapsoriasis patients were included, and no significant differences were reported between lesional and non-lesional skin.

Harkins et al. [80] evaluated the skin microbiome in four MF and two SS patients, and compared the results with 10 healthy volunteers. They analyzed viruses, fungi, and bacteria diversity. Among them, viruses and fungi did not show significant differences in abundance between lesional samples and healthy volunteers, as well as between MF and SS patients (Figure 1.). Despite earlier putative references to the role of *S. aureus*, the authors did not observe higher abundances in MF/SS patients compared to healthy volunteers. This study also did not show statistically significant differences in microbial diversity; however, higher relative abundances of *Corynebacterium* spp. and lower relative abundances of *Cutibacterium* spp. in CTCL patients were found (Figure 1). Moreover, *Corynebacterium tuberculostearicum* abundance was higher in stage IVA1 patients. Regarding bacterial diversity, the authors observed the greatest distinction between stage IVA1 patients and healthy volunteers, and suggested that bacterial shifts may correlate with disease stage or treatment status.

Dehner et al. [81] obtained skin swabs from seven MF patients (lesional and non-lesional skin) and compared them to samples of five healthy donors. Samples were collected from the arms, legs, and feet (Table 1.). The authors shed new light on the discussed topic, as they observed the presence of *Bacillus safensis*, a rare human skin commensal found only in individuals with diagnosed CTCL (Figure 1.). It should be mentioned that only skin swabs collected from lesional skin of the extremities (arms, legs) demonstrated the presence of *B. safensis*. However, *B. safensis* was not detected either in samples collected from the foot of CTCL patients or in the control group. The authors also analyzed data from previously performed studies by Salava et al. [76] and Harkins et al. [80]. During the analysis, they confirmed the presence of *Bacillus spp.*, which was in line with their results. Moreover, the authors obtained biopsies from two CTCL patients to investigate T-cell proliferation in response to patient-isolated bacteria. Therefore, they isolated, harvested, and seeded T-cells, and tested the cytokine concentration and proliferative response to *B. safensis*, *S. aureus*, *S. epidermidis*, *Deinococcus grandis*, *Acinetobacter radioresistens*, and *Staphylococcus cohni*. They observed that only *B. safensis* stimulated the proliferation of T-cells. Moreover, high levels of TNF-α, IFN-γ, IL-10, IL-17A, IL-21, and GM-CSF were observed [81]. These in vitro observations may suggest the putative role of *B. safensis* in T-cell activation and, thus, CTCL development, implying that microbial triggers initiate tumorigenesis.

So far, the largest study was introduced by Zhang et al. [82], who observed changes in the skin microbiome between different presentations of MF. In their study, samples were obtained from 39 patients from skin lesions and, as controls, non-lesional skin on the contralateral side (Table 1.). Again, no statistically significant differences were reported in the bacterial diversity and richness between lesional and non-lesional skin. However, similarly to Harkins et al., they observed higher *Corynebacterium* spp. abundance in lesional skin. Moreover, *Neisseriaceae* was more abundant in lesional skin, whereas non-lesional samples were characterized by an increased abundance of *Sandaracinobacter* spp. and *Enhydrobacter* spp. Furthermore, the authors observed microbiome alterations depending on the disease phenotype. In patients with marked erythema, an increase in *Staphylococcus* spp. was observed. This is in line with previous reports showing the role of *S. aureus* colonization in patients with the erythrodermic form of CTCL [59]. Thickened skin was characterized by a decrease in *Propionibacterium* spp. and *Bradyrhizobium* spp., and an increase in *Paracoccus* spp. Painful lesions were associated with decreased *Propionibacterium*, and increased *Bradyrhizobium* spp. and *Staphylococcus* spp, whereas excoriation was characterized by reduced *Conchiformibus* spp. Finally, pruritus was associated with an increase in *Sphingomonas* spp. and *Parvimonas* spp. (Figure 1).

Further in this section, we summarize the information about microorganisms linked to CTCL, and present the possible role of microbes described in the literature.

### 5.2. Staphylococcus aureus

The first report regarding the potential role of antigen persistence in CTCL appeared in the 70s [74]. *S. aureus* is a harmless commensal present on the healthy skin in 10–20% of people [83]. However, in patients with erythrodermic CTCL, a higher percentage of *S. aureus* skin and nostril colonization is observed. Alongside an impaired skin barrier, this usually causes subsequent bacterial infections [59,83,84,85]. Moreover, severe infections are the leading cause of death in advanced stages of CTCL, most of which are caused by *S. aureus*. Such modulations of the microenvironment contribute to the disease progression [59,83,86,87]. Investigating the role of *S. aureus* in CTCL patients, scientists focused on the relationship between *S. aureus* colonization, *S. aureus* toxins, and CTCL flares and/or the development.

There are two mechanisms of T-cell stimulation by bacterial superantigens—direct and indirect. The indirect mechanism assumes that non-malignant T-cells stimulated by the mentioned superantigens produce cytokines that activate malignant T-cells as the latter. The direct mechanism implies that superantigens stimulate malignant T-cells [88,89]. Particular *S. aureus* toxins, such as staphylococcal enterotoxins (SE), toxic shock syndrome toxin-1 (TSST-1), exfoliative toxins (ExT), and pore-forming alpha toxins, differently interact with T-cells. Most studies describe the role of SE in CTCL pathogenesis. SE triggers complex interactions between malignant and non-malignant T-cells, promoting broad T-cell activation in an MHC- and SE-dependent manner [90,91]. There are several mechanisms involved. Aside from MHC class II, FOXP3 expression and the IL2/STAT5 pathway are driven by SE [90]. Moreover, a link between SE-producing *S. aureus* and a high expression of oncogenic microRNA miR-155 has been shown [92]. Another example of malignant–non-malignant crosstalk involving SE is the activation of the STAT3/IL-10 axis that leads to the suppression of cellular immunity and anti-tumor responses [87]. All the above data confirm a substantial role of SE in many immunological pathways resulting in immune dysregulation in CTCL. When it comes to other *S. aureus* toxins, there are not much data available. The proliferation of Vβ-2-bearing malignant T-cells has been shown in response to ExT and TSST-1 [88,93]. Besides Vβ2 expression, some authors suggested increased Vβ5.1 usage in response to the stimulation of *S. aureus* superantigens. However, this observation was ambiguous [94,95,96]. Even though pore-forming alpha-toxin is expressed in 95% of *S. aureus* strains, limited data are available on its role in CTCL [97]. It seems that alpha-toxin shifts the balance in favor of malignant over non-malignant CD4+ T-cells. Malignant T-cells are resistant to the alpha-toxin effects in contrast to their non-malignant CD4+ counterparts [98]. Subsequent research showed that the population of CD8+ T-cells is very sensitive to alpha-toxin-induced toxicity, which leads to the depletion of non-malignant T-cells. Moreover, in most SS patients, alpha-toxin favors malignant T-cells and inhibits CD8+ T-cell-mediated anticancer immune responses, which enable the persistence and proliferation of malignant T-cells [99].

Despite numerous studies on the involvement of *S. aureus* and its toxins in CTCL pathogenesis, microbiome investigations did not confirm *S. aureus’s* role in the disease’s early stages. The available reports of *S. aureus* colonization concern the erythrodermic presentation of CTCL, which may suggest that these bacteria contribute to disease progression rather than CTCL development.

### 5.3. Cutavirus

The recently discovered Cutavirus (CuV), a member of the *Parvoviridae* family, was described for the first time in 2016. It was identified in fecal samples from patients with diarrhea. Then, CuV was detected in feces and skin samples of MF patients, and skin samples of parapsoriasis, eczema, melanoma, and skin carcinoma patients [100,101]. Vaïsänen et al. [102] showed the presence of CuV-DNA in the skin biopsies of 4 out of 25 patients with CTCL, and 4 out of 136 transplant recipients. The investigators did not observe CuV-DNA in healthy individuals. Interestingly, CuV-DNA was present not only in lesional skin, but also in non-lesional skin. This study also reported a significantly higher CuV-DNA prevalence in CTCL samples compared with transplant recipients and healthy adult samples. The authors implied a possible role of CuV in CTCL carcinogenesis [102]. On the other hand, Bergallo et al. analyzed 55 samples of CTCL patients, and did not observe the presence of CuV-DNA [103]. Furthermore, there are not enough data investigating the presence of CuV-DNA in the skin of CTCL patients. Even though Vaïsänen et al. [102] showed the prevalence of CuV-DNA in CTCL patients, this single report cannot be used to draw any further conclusions. Finally, the literature indicates that CuV may play a possible role in the process of carcinogenesis in patients with CTCL.

## 6. Therapeutic Approaches in CTCL and Their Influence on the Skin Microbiome

There are numerous management options for CTCL patients, including skin-directed therapy, systemic therapy, and others [104,105,106,107]. So far, no studies have analyzed changes in the skin microbiome before and after treatment in the same individual with CTCL. Zhang et al. [82] compared samples from patients on any therapy and those not yet receiving any treatment. Out of 39 patients, 12 were newly diagnosed. The remaining patients received multiple treatment modalities: 15 topical corticosteroids, three topical nitrogen mustard, two topical bexarotene, two topical imiquimod, four phototherapy, one radiotherapy, nine systemic therapy, and four adjuvant bleach baths. The authors identified a higher relative abundance of *Sarcina* spp. and a lower relative abundance of *Sphingomonas* spp. in the lesional skin after therapy.

### 6.1. Skin-Directed Therapy

The first-line treatment in CTCL’s early stages is skin-directed therapy, such as topical corticosteroid therapy (TCS) and phototherapy [106]. In MF patients, psoralen ultraviolet A (PUVA) and narrowband ultraviolet B (nb-UVB) are commonly used therapeutic options, and both methods are effective at producing a partial or complete response [108]. Most available data on the influence of mentioned therapeutic methods on the skin microbiome concerns AD. Kwon et al. [109] compared the impact of TCS and TCS combined with nb-UVB on the skin microbiome in AD. In both groups, the bacterial diversity of lesional skin increased after treatment. Interestingly, additional nb-UVB did not show a significant effect on the microbiome diversity, but 3 weeks after the discontinuation of phototherapy, a decrease in clinical severity scores and an increase in non-lesional skin microbiome were observed. In the TSC group, the severity scores increased, and changes in the non-lesional skin microbiome were not reported. Gonzalez et al. [110] also presented normalization of the skin microbiome after TCS in patients with AD. Moreover, a reduction of *S. aureus* colonization and an increased microbial diversity on the lesional skin were observed.

Regarding ultraviolet radiation (UVR), in AD patients, an increased diversity of the lesional skin microbiome was observed after 6-8 weeks of nb-UVB [111]. One study tested the role of UVA and UVB on the skin microbiome in healthy individuals. Burns et al. [112] presented that both types of UVR influence the skin microbiome composition. Unquestionably, both TCS and phototherapy influence bacterial diversity, causing its normalization. However, the discontinuation of treatment leads to disease flare and, hence, the development of dysbiosis.

### 6.2. Effect of Antibiotics on S. aureus Associated with CTCL

In patients with CTCL colonized with *S. aureus*, clinical improvement is associated with antibiotic treatment, resulting in *S. aureus* eradication [59,113,114]. Talpur et al. [59] observed the colonization of *S. aureus* in MF/SS patients on skin and nares. Four-to-eight weeks after the treatment (dicloxacillin, nafcillin, ampicillin, cefalexin, or clindamycin, depending on the sensitivity of the cultured bacteria), 30 out of 33 patients had negative skin cultures. Lindahl et al. [115] observed an improvement of the skin condition after intravenous antibiotic treatment (cephalosporins and metronidazole) in patients with stage IIB CTCL. What is more, they performed biopsies before and 2 months after treatment. Interestingly, besides clinical improvement, a decrease in cell proliferation and the expression of interleukin-2 receptor (IL2R)-a and tyrosine-phosphorylated STAT3 (pYSTAT3) was observed in immunohistochemical staining [116]. Due to the appearance of the new data, researchers expanded their study to assess bacteria on lesional skin in patients with advanced CTCL before, during, and after aggressive antibiotic treatment. During antibiotic treatment, the eradication of *S. aureus* was observed. However, in four out of six patients, the discontinuation of antibiotic therapy resulted in *S. aureus* recolonization. As the skin of CTCL patients is frequently colonized by methicillin-resistant *S. aureus*, long-term antibiotic treatment could promote resistant strains, causing even more problems with the future eradication of *S. aureus*. Hence, even if an improvement of the skin condition is observed, long-term antibiotic therapy should be reserved for a particular group of patients, and should not be used on a daily basis. Besides its antimicrobial effect, doxycycline has anti-inflammatory properties. Thus, El Sayed et al. [117] compared the therapeutic efficacy of this antibiotic and PUVA in early-stage MF in a randomized control trial. Additionally, the authors hoped to prove that apart from anti-inflammatory characteristics, doxycycline has an apoptotic effect on T-cells, previously seen in vitro [118]. However, in terms of partial response (ORR), reduction in composite assessment of index lesion severity (CAILS), modified severity weighted assessment tool (mSWAT), histopathology score, and CD3 expression, doxycycline proved to be less effective. This is another report showing that antibiotic treatment might slightly reduce the severity of skin lesions. However, it was associated with more expressed gastric side effects compared with PUVA, and should be used in a selected group of patients. Another report introduced the Duvic regiment, proposed to patients with erythrodermic CTCL, tumor stage, and *S. aureus* colonization. The mentioned treatment scheme consisted of intravenous vancomycin and cefepime in combination with antiseptic whirlpool baths and corticosteroids with alternating topical antibiotics; after which, a significant improvement of the skin condition was observed [119]. Besides antibiotic treatment, Lewis [120] suggested testing for bacterial colonization to identify individuals that should be treated with antibiotics. As genetic methods are still very expensive, standard bacterial cultures could be helpful to minimalize antibiotic use. Moreover, such proceedings allow susceptibility testing for antibiotics, helping to choose the right treatment option. Thus, patients with CTCL could benefit from new non-antibiotic therapies that eradicate bacteria colonization. In AD, bleach baths are a frequently used treatment option. Diluted sodium hypochlorite has antibacterial properties, causing oxidative injury and bacterial cell death [121]. Gonzales et al. [110] ascertained lower *S. aureus* abundance in AD patients using bleach baths, and confirmed earlier observations of skin condition improvement after *S. aureus* eradication. Therefore, studies examining the influence of bleach baths on the skin microbiome in other disease entities are needed.

## 7. Conclusions and Further Directions

There are still many unanswered questions regarding the role of the skin microbiome in CTCL. Firstly, as CTCL are a group of rare diseases, it takes time to recruit a representative group of patients. Existing studies included a small number of patients, mainly in early-stage CTCL. Thus, available data may serve as the basis for further investigations rather than the foundation for drawing conclusions.

Secondly, skin microbiome studies are challenging to conduct, as the skin microbiome depends on various factors. It differs between distinct niches, as well as between individuals.

The combination of these factors indicates a need for extensive studies, including sampling from different body parts. In addition, the performed skin microbiome investigations provide reasonable grounds for future multicenter, large population studies that will enable reliable conclusions on the role of microorganisms in CTCL development and/or progression.

There is an ongoing discussion on the relationship between dysbiosis and the development of skin diseases. External factors were alleged as the ones involved in the pathogenesis of CTCL. However, there are still questions considering microbiome interactions with the skin immune system, and whether dysbiosis is a causative factor or a consequence of the disease. Available microbiome reports emphasized the presence of *Bacillus safensis*, higher *Corynebacterium* spp. and *Neisseriaceae* abundance, and lower *Cutibacterium* spp. abundance on patients’ lesional skin. Moreover, it seems that various microbiome shifts characterize different MF presentations. Thus, these findings should be confirmed in subsequent skin microbiome studies.

In addition, existing skin microbiome investigations mainly focused on bacterial diversity. For example, only one study that included six patients mentioned viral and fungal investigations, and did not observe significant alterations. This indicates the need for future virome and mycobiome analyses in patients with CTCL.

Lastly, there are still no data on the influence of treatment on the skin microbiome in CTCL. A comparison of changes in the skin microbiome before and after treatment may help understand the pathogenesis, and could indicate which microorganisms are involved in disease exacerbation. Moreover, the identification of microbes may contribute to the improvement of future therapeutic management. Increasing microbial resistance to antibiotics is another concern. Thus, knowing the microorganisms responsible for disease exacerbation could help select the correct antimicrobial treatment. Therefore, non-antibiotic options affecting the skin microbiome, such as bleach baths, may be useful in patients with observed dysbiosis. That is why, besides investigations concerning the influence of currently available therapies on the skin microbiome in CTCL, studies on new non-antibiotic treatments restoring the microbiome balance could help in CTCL management.

## Figures and Tables

**Figure 1 pathogens-11-00935-f001:**
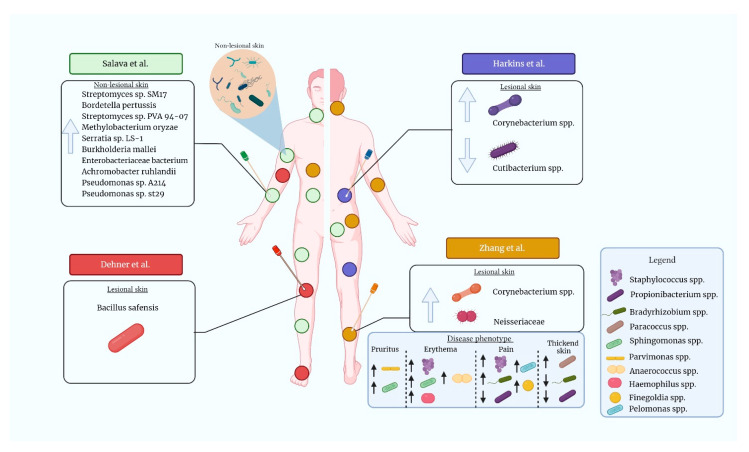
Sample sites and findings in CTCL microbiome studies. The colored circles represent sample sites in each microbiome study. Colors correspond to studies as follows: green—Salava et al. [76], blue—Harkins et al. [80], red—Dehner et al. [81], yellow—Zhang et al. [82] Only chosen findings on lesional and non-lesional skin are presented. The bacteria are shown only for illustrative purposes, as genetic sequencing methods have been used in the studies. This figure was created with BioRender.com.

**Table 1 pathogens-11-00935-t001:** An overview of methods used in recent CTCL microbiome studies.

Study	Cases	Controls	Sample Sites	CTCL Stage/Subtype	Methods	Therapy at Sampling
Salava et al. [76]	20 MF	healthy-appearing skin on the contralateral side of the body	extremities (5 thigh, 2 forearm, 1 upper arm, 1 shin); trunk (5 flank, 2 abdomen, 1 back, 1 buttock, 1 inguinal fold), 1 neck	IA-IIB	16S rRNA sequencing and WGS	11 bexarotene, 2 MTX, 6 no treatment
Harkins et al. [80]	4 MF and 2 SS (lesional and non-lesional skin)	10 healthy individuals (site-matched samples; age- and sex-matched individuals)	right and left lower back and bilateral posterior thighs	MF IA to IIIA SS IVA1	shotgun metagenomic sequencing	1 TCS, 1 TCS + PUVA, 1 TCS + photopheresis + IFα, 1 TS + bexarotene, 1 TS + MTX, 1 TCS + photopheresis + bexarotene
Dehner et al. [81]	7 MF	5 healthy individuals (body-site–matched skin samples); Non-lesional skin samples from MF patients (2 inches next to each matched lesion)	4 arm, 2 leg, 2 foot	MF IB, Follicular MF	16S rRNA sequencing	3 bexarotene + mechlorethamine, 4 no treatment
Zhang et al. [82]	39 MF	non-lesional skin in the contralateral side	14 trunk, 7 buttock, 14 extremities, 4 head and neck	I-IV	16s rRNA sequencing	12 no treatment, 15 TCS, 3 topical nitrogen mustard, 2 topical bexarotene, 2 topical imiquimod, 4 phototherapy, 1 RTH, 9 systemic therapy, 4 adjuvant bleach bath

CTCL—cutaneous T-cell lymphoma, MF—mycosis fungoides, SS—Sézary syndrome, WGS—whole-genome shotgun sequencing, TCS—topical corticosteroids, MTX—methotrexate, RTH—radiation therapy. No statistically significant differences between lesional and control samples were found regarding microbial diversity and richness.

## Data Availability

Not applicable.

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
