# Peer review of "The Skin Microbiome in Cutaneous T-Cell Lymphomas (CTCL)—A Narrative Review"

_pathogens, 2022, doi:10.3390/pathogens11080935_

Round 1
Reviewer 1 Report
Thank you for this nice review of the current available data on the microbiome of CTCL. The mentioning of Cutavirus is great, it's a promising topic in CTCL pathogenesis.
Author Response
Dear Reviewer,
We are very thankful for your precious time spent reviewing our manuscript. We appreciate your kind words and are very glad that you enjoyed our review, considering the role of the skin microbiome in CTCL.
Reviewer 2 Report
It is an interesting review discussing the link between skin microbiome and cutaneous T-cell lymphomas (CTCL). The review includes the following points:
a) Healthy skin microbiome. b)The role of the microbes in skin diseases development and progression. c)The skin microbiome and role of microorganisms in CTCL including mainly the roles of Staph aureus and Cutavirus. d)Therapeutic approaches in CTCL and their influence on the skin microbiome including Skin-directed therapy and antibiotics
e) Future directions
I have some suggestions
1- The review need careful language editing and to be revised by a native language speaker before resubmission
2- In the title "3. The role of the microbes in skin diseases development and progression". I suggest to make subtitles as 3.1 , 3.2 and each one include a specific skin disease as atopic dermatitis ,.. etc.
3- For Figure 1: is it designed by the authors? if yes, please mention the software/program used.
4- title :5.2. Antibiotics in CTCL:, since most literatures of this topic are focused or S. aureus infection. I suggest to rephrase it " Effect of antibiotics on S.aureus associated with CTCL.
5- Under the title 4; I suggest include the role of other bacteria such as S.epidermidis and its products such as cysteine protease and lantibiotics.
Author Response
Dear Reviewer,
We are very thankful for your precious time spent reviewing our manuscript and all your valuable comments and suggestions. A point-by-point response is provided in the attachment. You can find all changes in the updated manuscript version using the “Track Changes” function. Please see the attachment.

Round 2
Reviewer 2 Report
no further comments